# Seroprevalence of Crimean Congo Hemorrhagic Fever Virus in Occupational Settings: Systematic Review and Meta-Analysis

**DOI:** 10.3390/tropicalmed8090452

**Published:** 2023-09-19

**Authors:** Matteo Riccò, Antonio Baldassarre, Silvia Corrado, Marco Bottazzoli, Federico Marchesi

**Affiliations:** 1AUSL–IRCCS di Reggio Emilia, Servizio di Prevenzione e Sicurezza Negli Ambienti di Lavoro (SPSAL), Local Health Unit of Reggio Emilia, 42122 Reggio Emilia, Italy; 2Department of Experimental and Clinical Medicine, University of Florence, 50134 Florence, Italy; antonio.baldassarre@unifi.it; 3ASST Rhodense, Dipartimento della donna e Area Materno-Infantile, UOC Pediatria, 20024 Garbagnate Milanese, Italy; scorrado@asst-rhodense.it; 4Department of Otorhinolaryngology, APSS Trento, 38122 Trento, Italy; marco.bottazzoli@apss.tn.it; 5Department of Medicine and Surgery, University of Parma, 43126 Parma, Italy; federico.marchesi@unipr.it

**Keywords:** Crimean Congo Hemorrhagic Fever, epidemiology, CCHF, tick-borne pathogen, occupational settings

## Abstract

Crimean Congo Hemorrhagic Fever (CCHF) Virus can cause a serious human disease, with the case fatality ratio previously estimated to be 30–40%. Our study summarized seroprevalence data from occupational settings, focusing on the following occupational groups: animal handlers, abattoir workers, farmers, healthcare workers, veterinarians, rangers, and hunters. Systematic research was performed on three databases (PubMed, EMBASE, MedRxiv), and all studies reporting seroprevalence rates (IgG-positive status) for CCHF virus were retrieved and their results were reported, summarized, and compared. We identified a total of 33 articles, including a total of 20,195 samples, i.e., 13,197 workers from index occupational groups and 6998 individuals from the general population. Pooled seroprevalence rates ranged from 4.751% (95% confidence intervals (95% CI) 1.834 to 11.702) among animal handlers, to 3.403% (95% CI 2.44 to 3.932) for farmers, 2.737% (95% CI 0.896 to 8.054) among rangers and hunters, 1.900% (95% CI 0.738 to 4.808) for abattoir workers, and 0.644% (95% CI 0.223–1.849) for healthcare workers, with the lowest estimate found in veterinarians (0.283%, 95% CI 0.040–1.977). Seroprevalence rates for abattoir workers (odds ratio (OR) 4.198, 95% CI 1.060–16.464), animal handlers (OR 2.399, 95% CI 1.318–4.369), and farmers (OR 2.280, 95% CI 1.419 to 3.662) largely exceeded the official notification rates for CCHF in the general population. CCHF is reasonably underreported, and pooled estimates stress the importance of improving the adherence to personal protective equipment use and appropriate preventive habits.

## 1. Introduction

Crimean Congo Hemorrhagic Fever (CCHF) is a tick-borne illness caused by the CCHF Virus (CCHFV) [1], an enveloped RNA virus belonging to genus Orthonairovirus (family Nairoviridae, order Bunyavirales) [2,3]. Similar to other vector-borne pathogens, such as Tick-Borne Encephalitis Virus (TBEV) and West Nile Fever Virus (WNFV) [4,5,6], global distribution of CCHFV mirrors that of its reservoir host, the *Hyalomma* genus tick. Nowadays, CCHF is considered the most widespread viral tick-transmitted hemorrhagic fever, having been isolated throughout Africa, the Middle East, Southeast Asia, and Southern and Eastern Europe [1,7,8], as well as in Western and Northern Europe [1,9,10], as far north as Sweden [1,8,10,11].

CCHFV has been identified in a wide range of animal hosts, including birds (instrumental for the long-range expansion of its areal), hares, small rodents, and larger mammals such as sheep and buffalos [8,12], while humans are more properly considered casual hosts [1,8,12], being infected with CCHFV through tick bites, contact with blood and/or bodily fluids or tissues of a viremic animal or human, or handling and butchering of infected livestock [1,8,13,14]. Clinically, CCHF is characterized by the consequences of increased vascular permeability and cytokine storm [1,8,10,15,16], ranging from flu-like symptoms (fever, headache, myalgia, and malaise), photophobia, abdominal pain, diarrhea, and vomiting, to hemorrhagic manifestations that range from petechiae, epistaxis, and ecchymoses at venipuncture and injection sites, to severe internal hemorrhages, with a high case fatality ratio (CFR) [1,8,9,10,15,16,17,18].

Human occurrence of CCHF is considered rare: even though up to 3 billion people are considered globally at risk, no more than 15,000 infections are reported every year, with significant heterogeneity [5,6,8,17,19]. For instance, during 2018, a total of eight cases were officially reported for the whole of the European Union/European Economic Area (EU/EEA), while only four cases were reported during 2020 [5,6]. Nonetheless, since 2014, CCHF has been designated by the World Health Organization and then by the European Centre for Disease Prevention and Control (ECDC) as a high-priority emerging infectious disease for a number of reasons [5,6,8]. On the one hand, licensed vaccines and approved specific medical treatments are lacking [1,8,20]. On the other hand, CFR can exceed 30% or even >50% of incident cases depending on healthcare infrastructures [1]. In recent years, seroprevalence studies have indirectly suggested that the high CFR associated with CCHF may be a consequence of the likely underestimation of the actual global prevalence of CCHFV infections [1,8], due to a very high proportion of asymptomatic infections (>90%) [8]. In fact, a recent systematic review from Belobo et al. [8] reported a more cautious CFR estimate of 8.0% (95% confidence interval (95% CI) 1.0 to 18.9) in humans with recent CCFHV infection, and equal to 4.7% (95% CI 0.0 to 37.6) in individuals with past infections [1,8].

Due to the high prevalence in animal species, CCHFV has the potential to be acknowledged as a pathogen of occupational interest among workers occupationally exposed to animals, animal products, and animal bodies (e.g., abattoir workers, animal farmers, rangers/hunters, veterinarians, etc.) [12,13,21,22,23,24,25], but also in subjects occupationally exposed to contaminated bodily fluids (i.e., healthcare workers) [12], as otherwise suggested by previous reports on small nosocomial outbreaks [26,27,28]. Even though working in direct contact with infected animals and animal products has been repeatedly associated with increased risk for CCHFV infection, and despite that several systematic reviews on CCHFV in humans have been published [8,12], including some estimates on the seroprevalence rate among healthcare workers [8,10,21], to the best of our knowledge, no comprehensive summary of seroepidemiological studies from occupational settings has ever been published. Therefore, we designed the present systematic review with meta-analysis in order to provide a summary of extant literature on CCHFV seroepidemiology in occupational settings. The collected evidence would serve as a basis to guide priorities in focusing prevention efforts according to the different geographic areas and specificities of occupational health frameworks, potentially enabling competent health officers and occupational physicians (where implemented by the underlying legal framework) to provide appropriate prevention recommendations for workers and employers.

## 2. Materials and Methods

A systematic review was designed in accordance with the updated PRISMA (Prepared Items for Systematic Reviews and Meta-Analyses) guidelines [29,30] and registered in the International Prospective Register of Systematic Reviews, or PROSPERO, with the progressive registration number CRD42023448537 (Appendix A).

### 2.1. Research Concept

The research question was defined in accordance with the “PICO” (patient/population/problem, investigated result, control/comparator, outcome) strategy (Appendix B, Table A1). More precisely, we specifically targeted groups of workers potentially exposed to tick bites (P) and retrieved seroprevalence rates for previous exposure to the CCHF virus (i.e., CCHFV-specific IgG) (I). Their estimates were compared to healthy individuals not occupationally exposed to ticks and tick bites and, therefore, not reasonably exposed to CCHFV (C), to ascertain whether occupationally exposed individuals exhibited an increased risk for developing CCHFV infection (O).

### 2.2. Research Strategy

From 1 to 15 July 2023, the scientific databases PubMed and EMBASE and the preprint repository MedRxiv were searched without any backward chronological restriction. The research strategy resulted from the combination of the following search strings, respectively, for PubMed (through Medical Subject Heading (MeSH) terms) and EMBASE:(a)(“Crimean Congo hemorrhagic fever” OR “Crimea Congo hemorrhagic fever” OR “Congo Crimea hemorrhagic fever” OR “Crimean Congo hemorrhagic fever virus” OR “Crimean Congo” OR “Crimea-Congo”) AND (“occupation*” OR “work related” OR “worker*”) AND (“seroprevalence” OR “epidemiology”).(b)(‘crimean congo hemorrhagic fever’ OR ‘crimean congo hemorrhagic fever’/exp OR ‘nairovirus’) AND (‘occupation’ OR ‘work’ OR ‘workforce’) AND (‘seroprevalence’ OR ‘epidemiology’ OR ‘prevalence’).

Documents eligible for review were original studies, including seroprevalence studies designed as cohort, case-control, and cross-sectional studies. Case series, case reports, and retrospective reports from local and national registries were excluded from the eventual estimates. Suitable publications were either obtained through online repositories online or through inter-library loan, and were excluded if: (1) full text was not available, (2) the main text was written in a language not directly understood by the reviewers (i.e., English, Italian, German, French, Spanish, and Farsi), (3) reports lacked an appropriate or only vaguely defined geographical setting, (4) estimates other than IgG prevalence were provided, and (5) job titles of the exposed workers were not provided or only vaguely defined (e.g., “rural activities”, “interaction with animals”, etc.).

### 2.3. Screening

According to the PRISMA guidelines [29,30], all retrieved items were initially title screened for their relevance to the subject, with subsequent analysis of their abstracts. If considered in line with the aims of the present review, corresponding full-text versions were assessed. All items were independently rated by two investigators (A.B. and F.M.), and disagreements were either resolved by consensus or, where consensus was not reached, through input from the chief investigator (M.R.).

### 2.4. Summary of Retrieved Data

The data included:(a)Settings of the study: country, region, and prevalence year.(b)Characteristics of the serologic assay.(c)Occupational groups of the sampled cases: For the aims of this review, we considered the following occupational groups, previously associated with a higher risk of CCHFV infection: (1) animal handlers—all workers whose job is to be in charge of and control animals, including, but not limited to, livestock farmers, shepherds, milkmen, etc.; (2) abattoir workers—all workers involved in meat processing, at industrial and non-industrial levels; (3) farmers—a person who owns or manages a farm, with the exclusion of livestock farmers who were included among animal handlers; (4) healthcare workers (HCWs)—all professionals working in a hospital and/or a care center, irrespective of the specific job title; (5) veterinarians—professionals practicing veterinary medicine or surgery, qualified for treating sick or injured animals and for conducting their health assessments; (6) rangers—persons whose job is to look after forests or large parks; (7) hunters—individuals hunting wild animals for food or as a sport, not otherwise included in the previous occupational groups.(d)Total number of prevalent cases by occupational group. If only collective estimates were provided the study was removed from the analysis.(e)Number and characteristics of the reference population (if provided).

### 2.5. Risk of Bias Analysis

In order to assess the potential risk of bias of the retrieved studies, a specific analysis was performed by means of the risk of bias (ROB) tool provided by the National Toxicology Program’s (NTP) Office of Health Assessment and Translation (OHAT) [31,32]. In its current version, the OHAT ROB tool evaluates the internal validity of a study through the analysis of six possible sources of bias (i.e., participant selection, confounding, attrition/exclusion, detection, selective reporting, and other sources). The aim of the ROB analysis is to ascertain whether any of the aforementioned dimensions were likely to compromise the credibility of the link between exposure and outcome, with potential answers ranging from “definitely low”, “probably low”, “probably high”, to “definitely high”. By its design, the OHAT ROB tool does not apply an overall rating for each study, stressing that even studies with “probably high” or “definitely high” ratings should be retained in the pooled analyses.

### 2.6. Data Analysis

All estimates for occupational groups from the included studies were initially summarized through a descriptive analysis, with subsequent calculation of crude prevalence figures as per 100 population. If a study did not include raw data, either as prevalent cases or a referent population, such figures were reverse-calculated from the available data. Risk ratios (RR) for the seropositive status of each occupational group and their corresponding 95% confidence intervals (95% CI) were calculated in a bivariate analysis by assuming the seropositive status retrieved from the general population as a reference group.

Pooled seroprevalence estimates as well as pooled odds ratios (OR) and their corresponding 95% CI compared to the general population were calculated for a meta-analysis of the retrieved studies. A random-effect model was preferred over a fixed-effect model in order to cope with the presumptive heterogeneity in the study design. The inconsistency between the included studies was defined as the percentage of total variation across studies likely due to heterogeneity rather than chance [33], and was quantified by means of the I^2^ statistic, categorized as follows: I^2^ ranging from 0 to 25%, low heterogeneity; I^2^ ranging from 26% to 50%, moderate heterogeneity; I^2^ ≥ 50%, substantial heterogeneity. In order to cope with the potential small size of the meta-analysis performed for the occupational groups, as previously suggested by Von Hippel [33], the 95% CI values of each I^2^ estimate were provided. Sensitivity analysis was performed to evaluate the effect of each study on the pooled estimates by the exclusion of one study at a time. Any significant change in pooled estimates was reported.

Potential publication bias was ascertained through calculation of contour-enhanced funnel plots, and their asymmetry was eventually assessed by means of the Egger test statistic. Small study bias was eventually assessed by generating corresponding radial plots. The level of evidence was applied to the GRADE criteria and reported [20,34,35].

All calculations were performed in R (version 4.3.1) [36] and RStudio (version 2023.06.0 Build 421; RStudio, PBC; Boston, MA, USA) software by means of the packages meta (version 6.5.0) and fmsb (version 0.7.5). A PRISMA2020 flow diagram was designed by means of the PRISMA2020 package [37].

## 3. Results

### 3.1. Descriptive Analysis

As shown in the flow chart reported in Figure 1, a total pool of 378 entries was retrieved from database searches (i.e., 108 from PubMed, 184 from MedRxiv, and 86 from EMBASE).

Out of the total pool, 259 (68.52%) papers were identified as duplicate entries and were removed. Screening by title and abstract of the remaining 119 records (31.48% of the original pool) led to the removal of 65 further articles (17.20%). A total of 54 entries were eventually assessed and their full text was reviewed (14.29%); of these, 28 were further removed from the analyses due to not meeting the inclusion criteria (7.41%). Citation searching from the eventual sample of 26 papers obtained through database searches (6.63% of the initial sample) identified 13 additional records. Of these, 5 were removed for not fulfilling the inclusion criteria. Therefore, qualitative and quantitative analyses were performed on a final pool of 33 papers.

The 33 studies included in the analyses, all of them published from 2007 to 2022, are summarized in Table 1 [13,14,21,22,23,24,25,38,39,40,41,42,43,44,45,46,47,48,49,50,51,52,53,54,55,56,57,58,59,60,61]. Among them, 17 (51.51%) were published after 2014, when the WHO first identified CCHF as a global health threat [20]. All studies were based on enzyme-linked immunoassay (ELISA) and included a total of 20,195 samples: 13,197 (65.35%) were associated with occupational groups, while 6698 samples were considered from the general population (34.65%).

The majority of studies (10 out of 33, 30.30%) reported data from Turkey [21,22,43,44,47,53,56,58,60,63], for a total of 5186 subjects (20.43% of the total sample, range 75 to 2319 patients per paper), followed by Iran (5 studies, 15.15%) for a total 760 subjects (3.76% of the total sample, range 104 to 250) [23,39,46,52,59], and Greece (3 studies including a total of 2095 subjects, 10.37% of the total sample, range 207–1611) [40,41,62]. Pakistan (1505 subjects, 7.45% of the total sample) [48,49] and South Africa (1427 subjects, 7.07% of the total sample) [25,45] provided 2 studies each (6.06%). One single report was provided from India (No. = 4953, 24.53%) [55], Madagascar (No. = 1995, 9.88%) [38], Mainland China (No. = 1657, 8.21%) [57], Kazakhstan (No. = 946, 4.68%) [51], Uganda (No. = 800, 3.96%) [54], Spain (No. = 516, 2.56%) [50], Tunisia (No. 219, 1.08%) [42], Ghana (No. = 109, 0.54%) [13], Myanmar (No. = 102, 0.51%) [61], Malaysia (No. = 85, 0.42%) [24], and Saudi Arabia (No. = 80, 0.40%) [14].

The eventual sample included: 17 estimates based on abattoir workers for 3657 total samples (range 6 to 1995) [13,14,23,25,38,39,40,41,42,45,46,47,48,50,52,55,59], 14 estimates based on farmers for 4558 samples (range 12 to 1034) [24,25,40,41,43,45,49,51,53,54,55,57,58,62], 12 estimates based on animal handlers for 3505 samples (range 32 to 768) [40,44,45,47,48,50,51,53,54,55,57,60], 8 estimates based on veterinarians (354 samples, range 1 to 117) [25,45,47,50,51,52,55,56], and 7 estimates based on HCWs (765 samples, range 10 to 307) [21,22,46,50,51,55,63]. Due to the reduced number of available estimates, rangers and hunters were collapsed into a single group that included 8 estimates (range 2 to 72) for a total of 328 samples [25,40,41,43,45,46,50,61]. Moreover, 12 studies provided estimates for the general population of the assessed areas [42,43,46,47,49,50,51,53,55,61,62], whose size ranged between 19 to 1677 samples.

As shown in Table 2, the crude seroprevalence for CCHFV was highest among animal handlers (9.22%), with individual estimates ranging between no positive case [45] and 37.82% in the report from Atim et al. [54], followed by farmers (5.67%). Again, individual estimates ranged between 0.00% [24,45,51], 0.10% [55], and well over 10% [40,53], peaking at 18.12% in the study of Atim et al. [44]. When dealing with abattoir workers, crude prevalence was estimated at 2.24%, with individual estimates that ranged from three negative reports [14,25,41] to 10.94% in the study by Salmanzadeh et al. [52], 14.81% in the report by Chinikar et al. [59], and 16.49% from the study of Mostafavi et al. [23]. Lower prevalence rates were calculated for HCWs (0.57%), and particularly among veterinarians (0.28%), as only 1 positive case was reported across the 7 studies and in the 354 sampled professionals. Regarding rangers and hunters, a total of 13 positive cases out of 328 samples were identified, with a crude prevalence of 3.39%. However, more than half of the reported cases were included in the single report from Myanmar (prevalence of 10.77%) [61]. The remaining individual estimates encompassed three negative reports [43,45,52], three reports ranging between 3.13% [41], 3.45% [40], and 4.17% [25], as well as an outlier represented by the study of Arteaga et al. [50], which only included two cases, one of which was IgG-positive for CCHFV (50.00%).

In the general population, a total of 192 positive cases were identified among the 6998 sampled individuals, for a crude positive rate of 2.74%. Consequently, when assuming the general population as the reference group, the highest risk for CCHFV seropositivity was associated with animal handlers (RR 3.359, 95% CI 2.823 to 3.997), followed by farmers (RR 2.079, 95% CI 1.732 to 2.496), while HCWs and veterinarians exhibited a substantially reduced risk (RR 0.238, 95% CI 0.098 to 0.577, and RR 0.103, 95% CI 0.015 to 0.733, respectively). Interestingly, for the remaining occupational groups, no substantial differences were reported compared to the general population for both abattoir workers (i.e., RR 0.817, 95% CI 0.633 to 1.533) and rangers and hunters (RR 1.445, 95% CI 0.833 to 2.506).

### 3.2. Risk of Bias

A summary of the risk of bias (ROB) assessment for the retrieved studies is reported in Figure 2, while details on single studies are included in Appendix B, Table A2.

Briefly, majority of reports were either of good or very good quality, as the study design was developed to cope with potential selection bias (D1), exposure assessment (D2), and outcome assessment (D3). Additionally, reporting bias (D5: the elective inclusion of outcomes in the publication of the study on the basis of the results) was unlikely in the included studies. In fact, the main issues were associated with the identification and handling of confounding factors (D4), as potential sources of tick bites and non-occupational exposures to CCHFV were not properly investigated in 4 out of 33 studies (12.12%) [22,23,46,52]. Moreover (D6: other bias), a substantial share of studies (8 out of 25 studies, 32.00%) were likely affected by some uncertainties in the assessment of the individual habits, attitudes, and knowledge of participants [24,38,42], as well as in the detailed description of data handling and statistical analysis [22,23], or reporting in the randomization strategy [39,47,52].

### 3.3. Meta-Analysis

Pooled seroprevalence rates for CCHFV were estimated through a random-effect model meta-analysis. Estimates for individual groups are provided in Table 3, while corresponding forest plots are included in Appendix B, Figure A1, Figure A2, Figure A3, Figure A4, Figure A5 and Figure A6.

Estimates were 4.751% (95% CI 1.834 to 11.702) among animal handlers, 3.403% (95% CI 2.944 to 3.932) for farmers, 2.737% (95% CI 0.896 to 8.054) among rangers and hunters, 1.900% (95% CI 0.738 to 4.808) for abattoir workers, and 0.644% (95% CI 0.223 to 1.849) for HCWs, with the lowest estimate found in veterinarians (0.283%, 95% CI 0.040 to 1.977). Half of the estimates were affected by substantial heterogeneity; more precisely, animal handlers (I^2^ = 96.0%, 95% CI 94.4% to 97.1%; Q = 273.80, *p* < 0.001), farmers (I^2^ = 91.1%, 95% CI 86.8% to 94.0%; Q = 146.16, *p* < 0.001), and abattoir workers (I^2^ = 77.5%, 95% CI 64.4% to 85.8%; Q = 71.19, *p* < 0.001). However, when focusing on the 95% CI of I^2^ estimates for healthcare workers, rangers/hunters, and veterinarians, upper limits also exceeded 50.0%, stressing a potential bias due to the small number of studies.

Seroprevalence rates for occupational groups were compared to the general population when available, and pooled OR with corresponding 95% CI are provided in Table 4 and Appendix B, Figure A7. Abattoir workers were characterized by a higher likelihood of CCHFV seropositive status (OR 4.198, 95% CI 1.060 to 16.464), followed by animal handlers (OR 2.399, 95% CI 1.318 to 4.369) and farmers (OR 2.280, 95% CI 1.419 to 3.662). On the contrary, no substantial differences were reported for veterinarians (OR 7.966, 95% CI 0.261 to 242.834), rangers and hunters (OR 4.115, 95% CI 0.110 to 153.426), and HCWs (OR 3.678, 95% CI 0.620 to 21.835).

The heterogeneity of individual estimates was suggested as low for healthcare workers (I^2^ = 0.0%, 95% CI 0.0% to 74.6%, *p* = 0.967) and abattoir workers (I^2^ = 5.9%, 95% CI 0.0 to 85.6%, *p* = 0.363), moderate for animal handlers (I^2^ = 38.4%, 95% CI 0.0% to 74.1%, *p* = 0.136), and substantial for veterinarians (I^2^ = 55.0%, 95% CI 0.0% to 89.0%, *p* = 0.136), farmers (I^2^ = 55.8%, 95% CI 6.6% to 79.0%, *p* = 0.021), and rangers/hunters (I^2^ = 78.6%, 95% CI 31.2% to 93.3%, *p* = 0.009). However, when taking into account the upper limits, all assessed occupational groups were characterized by estimates exceeding 70%, suggesting that the groups characterized by seemingly low point estimates for heterogeneity were otherwise biased by the small size of the meta-analysis.

### 3.4. Sensitivity Analysis

Sensitivity analysis was performed by removing a single study at a time. Pooled estimates for prevalence (Appendix B, Figure A8) were not affected in terms of residual heterogeneity, which remained substantial for animal handlers, farmers, and abattoir workers. On the contrary, when the seroprevalence rates of the occupational groups were compared to the general population (Appendix B, Figure A9), estimates for animal handlers were affected by the removal of the studies by Çitil et al. [53] (OR 2.39, 95% CI 1.32 to 4.33, I^2^ = 9%) and Yagci-Caglayik et al. (OR 1.99, 95% CI 1.17 to 3.38, I^2^ = 18%) [60]. When dealing with abattoir workers, the removal of the study by Wasfi et al. [42] led to an estimated OR equal to 9.04, 95% CI 1.40 to 58.22, with no residual heterogeneity. Interestingly, the removal of the study by Gazi et al. [43] had a similar impact on the pooled estimate for rangers and hunters (OR 243.0, 95% CI 8.13 to 7260.08), and that by Head et al. on healthcare workers (OR 54.11, 95% CI 1.51 to 1942.31) [51].

### 3.5. Analysis of Publication Bias and Small Study Bias

Potential publication bias is assessed in Figure 3 by analysis of funnel plots. Funnel plots are a graphical representation of the sample size, or an index of precision (*Y*-axis) plotted against the effect size they report (*X*-axis) [30].

As the size of the sample increases, estimates are likely to converge around the true underlying effect size. Consequently, if the analysis is not affected by publication bias, an even scattering of point estimates is expected. On the contrary, when publication bias has occurred, an asymmetry in the scatter of small studies can be spotted, with more studies showing a positive result than those showing a negative result. Visual inspection of contour-enhanced funnel plots suggested substantial evidence of publication bias for most of the included groups, which was confirmed by Egger’s test (i.e., the linear regression analysis of the intervention effect estimates on their standard errors weighted by their inverse variance) for abattoir workers (Figure 3a, *p* < 0.001) and farmers (Figure 3c, *p* = 0.027). A likely publication bias was similarly suggested for studies based on animal handlers (Figure 3b, *p* = 0.078), while it reasonably spared HCWs (Figure 3e, *p* = 0.602), veterinarians (Figure 3f, *p* = 0.234), and rangers/hunters (Figure 3d, *p* = 0.218) (Table 5).

Radial plots were calculated in order to ascertain the small study bias, and the corresponding graphs are reported in Figure 4. According to Galbraith [30,32], radial plots are produced by first calculating the standardized estimates or z-statistics by dividing each estimate by its standard error (SE), then scatter plotting each z-statistic (*Y*-axis) against 1/SE (*X*-axis). On the one hand, it is expected that larger studies (characterized by a smaller SE and a larger 1/SE) will be observed to aggregate away from the origin. On the other hand, small study bias is usually ruled out by even plotting of estimates along the regression line. Visual inspection suggested that small study bias can be reasonably ruled out among abattoir workers (Figure 4a), animal handlers (Figure 4b), and HCWs (Figure 4e), while it reasonably affected all other estimates.

### 3.6. Summary of Evidence

The level of evidence by GRADE for prevalence estimates (Appendix B, Table A3) was moderate for abattoir workers, farmers, animal handlers, and veterinarians, due to the findings being downgraded based on the quality of the source studies, with low evidence for estimates on rangers/hunters and healthcare workers. When accounting for the corresponding estimates based on the general population (Appendix B, Table A4), the evidence was again moderate for animal handlers and farmers, as well as for healthcare workers, while it was low for abattoir workers, and very low for rangers/hunters and veterinarians due to sample size and quality issues.

## 4. Discussion

In the present systematic review and meta-analysis, the pooled seroprevalence for CCHFV-IgG on occupationally exposed individuals in the timeframe of 2006–2022 ranged between 0.283 per 100 people (95% CI 0.040 to 1.977) among veterinarians, 0.644 per 100 people (95% CI 0.223 to 1.849) among HCWs, 1.900 per 100 people (95% CI 0.738 to 4.808) for abattoir workers, 2.737 per 100 people (95% CI 0.896 to 8.054) among rangers and hunters, 3.403 per 100 people (95% CI 2.944 to 3.932) for farmers, and 4.751 per 100 people (95% CI 1.834 to 11.704) among animal handlers. The overall quality of the estimates was partially or even substantially impaired by quality issues of the studies and their limited sample sizes, but the sensitivity analysis did not recognize any substantial effect of single estimates on the pooled ones (Appendix B, Table A3 and Table A4).

Compared to the general population, the risk for seropositivity was greater among abattoir workers (OR 4.198, 95% CI 1.060 to 16.464), followed by animal handlers (OR 2.399, 95% CI 1.318 to 4.369) and farmers (OR 2.280, 95% CI 1.419 to 3.662). Again, studies were affected by quality issues and sample sizes, as the removal of the studies by Çitil et al. [53] and Yagci-Caglayik et al. [60] substantially affected the pooled estimates for animal handlers, with a similar impact of the removal of the study by Wasfi et al. [42] on animal handlers. Even though rangers and hunters (OR 4.115, 95% CI 0.110 to 153.426), healthcare workers (OR 3.678, 95% CI 0.620 to 21.835), and veterinarians (OR 7.966, 95% CI 0.261 to 242.834) seemingly exhibited seropositive rates similar to those found among the general population, pooled estimates for rangers and hunters and healthcare workers were substantially affected by the removal of a single study each, namely that by Gazi et al. [43] for the former group, and that by Head et al. [51] for the latter. In other words, the present study suggests that a substantial share of workers involved in agricultural tasks and the meat-processing industry are exposed to this pathogen, and that their risk is substantially greater than that for the general population [7,8,10,15,17]. Consequently, our results are of substantial interest for public health professionals and particularly for occupational physicians, for several reasons.

Despite the increasing claims for the potential health threat associated with CCHFV infection [5,6], this pathogen often remains underestimated, not only by the general population, but also among healthcare professionals [64,65]. Nonetheless, pooled data stress how CCHFV infection may represent a far more common occurrence, even in countries and settings not usually associated with CCHF. According to the available seroprevalence studies, CCHFV should be understood as more frequently reported among adults from Europe, Eastern Mediterranean, upper-middle-income, and high-income countries [1,7,18], but our review was also able to collect some evidence from Sub-Saharan Africa [25,38,45], Mainland China [57], and Southeast Asia [24,61]. In fact, it is reasonable that notification rates only represent the tip of a very larger burden of disease, particularly in occupational settings [9,10,15,50]. For example, focusing on European Union countries included in the pooled analyses, a seropositive status was identified in 3 out of 516 samples from the study of Arteaga et al. (0.58%) [50], in 61 out of 1611 samples from a nationwide study by Sidira et al. (3.79%) [62], in 6 out of 277 individual samples by Sidira et al. (2.16%) [41], and in 14 out of 207 samples by Sargianou et al. (6.76%) [40], with the former study being from Spain, while the other 3 were from Greece. On the contrary, official figures provided by the ECDC for the timeframe of 2008–2021 report a total of 9 cases of CCHF for Spain, and 2 cases of CCHF in Greece, with incidence rates that consistently remained well below 0.1 cases per 100,000 people [5,6]. As recently stressed by reports from hyperendemic areas [10,17,54], including the occupational report by Atim et al. [54], a radical paradigm shift is, therefore, conceivable when pondering the CCHF-associated actual burden of disease. Similar to other vector-borne diseases such as West Nile Fever [66,67,68,69,70,71,72], Dengue [73,74,75,76], and Tick-Borne Encephalitis [77,78,79,80,81], CCHF should be understood as a relatively rare occurrence following CCHFV infection, while majority of cases either go unnoticed or misunderstood as “summer flu” [82]. Far from dismissing the potential significance of this pathogen, a more appropriate appraisal of CCHFV in the evolving landscape of vector-borne diseases underlines the role of occupational physicians and medical surveillance in occupational settings as instrumental in improving our appropriate understanding of the CCHF epidemiology [83,84,85]. Where implemented by local legal framework, occupational physicians are medical professionals actively involved in health promotion in the workplace [86,87,88], also being involved in the communication of risk-appropriate preventive measures [88,89,90], potentially participating in the adaptation of workplaces’ design in accordance with appropriate health and safety requirements [85,88,91,92]. Therefore, raising their awareness on vector-borne pathogens such as CCHFV could not only improve the actual surveillance system, but could also provide the first step of an integrated preventive strategy, including the characterization of individual risk factors that may turn a mostly indolent infection into a potentially life-threatening clinical syndrome.

The status of CCHFV in occupational settings could, therefore, be acknowledged as quite similar to other zoonotic pathogens of occupational interest, e.g., Hantaviruses, Tick-Borne Encephalitis Virus (TBEV), *Borrelia burgdoferi* (the causal agent of Lyme disease), and *Coxiella burnetii* (the causal agent of Q fever) [91,93,94,95]. The strong association of seroprevalence for CCHFV with the meat industry (more precisely, for abattoir personnel), and tasks characterized by strong interactions between personnel and animals, collectively stress the importance for implementing appropriate preventive measures and consistent medical surveillance for biological risk agents, at least in settings characterized by interaction with potentially infected animals and ticks, and/or animal blood and bodily fluids [5,6,13,23,52,59]. Not coincidentally, although rangers and hunters deliberately interact with environmental settings characterized by a higher risk of tick bites, their pooled risk for CCHFV seropositivity was substantially superimposable to the risk of the general population. Therefore, as previously stressed by several health authorities [5,6,20,96], pooled prevalence rates ultimately stress the importance of personal protective equipment (PPE) in abattoir workers, while the estimates for CCHFV seroprevalence among farmers and animal handlers pinpoints the role of behavioral preventive interventions against ticks and their bites. Professionals working in at-risk areas and performing agricultural activities should implement a series of measures that have been proven effective in reducing the risk of tick bites (e.g., use of tick repellents such as DEET, wearing clothes that minimize skin exposure) and the subsequent spreading of other tick-borne pathogens, such as TBEV and *Borrelia burgdorferi* [5,6,20,96,97]. Even though there is no clear evidence that early tick head removal could radically decrease the risk for contracting CCHFV and ultimately developing CCFH, at-risk workers should be taught how to perform a self-check for ticks and how to properly remove an attached tick [96]. Eventually, from a “One Health Perspective”, public health professionals should promote the application of acaricides on livestock, an intervention that is particularly useful to protect not only animal handlers in agricultural settings, but also in slaughterhouses, and that also reduces the risk for transporting CCHFV-infected ticks to other regions through transport of animals. In this regard, it is reasonable to speculate that the low seroprevalence we reported among veterinarians may be associated with the higher adherence to preventive measures usually associated with those professionals [98].

Another issue raised by our data is the potential occurrence of CCHFV infection among HCWs [12,21,99]. Some previous reports, including two systematic reviews, specifically including cases of nosocomial transmission [12,27], suggested that CCHFV may be quite distinctive when compared to other arboviruses in being possibly associated with a higher likelihood of healthcare-associated infections [16,21,97,99]. However, our data suggest that the actual risk for HCWs would be similar to individuals from the general population [16,21,22,46,55,97,99]. Hence, the actual occupational status of most CCHFV infections among HCWs could be questioned, even in regions associated with high prevalence of this pathogen. The occupational status of CCHFV infections should be limited to cases where an index case is identified, and potential contacts are properly tracked.

### Limits

Despite its potential significance, even from the real-world perspective of occupational physicians, our study was affected by several significant shortcomings.

First, even though the studies we retrieved and included in the analyses were mostly of appropriate quality (as summarized by the ROB tools), some significant shortcomings in their design should be addressed. More precisely, a substantial share of studies did not address potential co-exposures and non-occupational risk factors for tick bites and/or CCHFV infection [45,46,48,49,52,55]. Even though this information gap does not impair the reliability of the collected results per se, it reduces our capability to appreciate the share of infections that have likely occurred because of the interaction with blood and bodily fluids, particularly in the meat-processing industry.

Second, even though nearly all studies extensively reported how the samples were recruited, the actual representativity of the assessed occupational groups from the targeted areas remains unclear. This issue is particularly significant when dealing with estimates from agricultural settings and/or low-income countries, where occupational and residential environments are hardly dichotomized [93,94], and job descriptions could fail to appreciate the actual exposures [93,94,100,101,102,103]. As participants can perform several tasks at the same time (e.g., agricultural workers that also care for animals on a daily basis, possibly performing their slaughtering), corresponding sources of exposure are also highly overlapped, suggesting a quite cautious appraisal of the eventual estimates.

Third, the collected studies were quite heterogenous in geographical terms, sample size, and sampling strategy. Moreover, even in the very same parent country, substantial socioeconomic differences among the sampled regions could impair the representativity of the pooled results. For instance, the pooled prevalence among farmers ranged between 0.10 per 100 workers (95% CI 0.00 to 0.54) in India [55], to 5.53 per 100 workers (95% CI 3.06 to 9.08) in Greece [40,41,62], 9.71 per 100 workers (95% CI 4.51 to 19.68) in Turkey [43,53,58], and 18.12 per 100 workers (14.52 to 22.17) in Uganda [54]. The very high pooled estimate for Turkey was based on three substantially different estimates (*p* < 0.001), more precisely: the study by Gazi et al. [43], reporting on the economically developed area of Manisa (western Anatolia), with a prevalence of 3.88 per 100 workers (95% CI 1.79–7.25) [43], the report of Çitil et al. [53] from the agricultural region of Tokat (Northern Anatolia), with an estimated prevalence of 10.88 per 100 workers (95% CI 1.79 to 7.25) [53], and that from Ertogrul et al. [58], with a prevalence of 19.76% (95% CI 14.01 to 26.62) in 167 total farmers from Aydin in a rural area of Western Anatolia. Similarly, the pooled estimate for abattoir workers included 5 estimates from various regions in Iran, with pooled estimates ranging from 2.00 per 100 workers (95% CI 0.05 to 10.65) [46], to 4.41 (95% CI 0.92 to 12.36) [39], 10.94 (95% CI 4.51 to 21.25) [52], 14.81 (95% CI 8.71 to 22.94) [59], and peaking at 16.49 per 100 workers in the report by Mostafavi et al. [23], and this difference was properly captured by substantial heterogeneity (I^2^ = 61% for subgroup analysis).

Fourth, even though no backward limit was defined, the collected studies were performed across a very broad timespan, from 2006 to 2022, with some studies beginning sample collection in 2003–2004 [21,59], or even in the previous decades. Laboratory techniques have evolved over time, and even though ELISA and immunofluorescence assays have replaced older techniques [2], no commercial test has been approved for routine human diagnostics [2,3,104]. While the overall diagnostic performances of commonly referred tests are far from optimal (sensitivity 80.4%, 95% CI 69.5 to 91.3, specificity 100%) [2], some warnings have been issued about the potential sensitivity of genetic variants of CCHF, such as the strain AP92 [41]. Nonetheless, no significant time trend was identified (Appendix B, Figure A10): despite some claims about increasing seroprevalence rates because of increased testing capacity [2,8,17], our estimates were seemingly not affected by any decennial trend.

Finally, our study was reasonably affected not only by publication bias, but also small study bias. This is particularly significant when taking into account that some studies, for example the reports by Arteaga et al. [50], Aydin et al. [47], and Vawda et al. [45], included several occupational groups of limited size, often less than 50 individuals, and that all “a priori” assumptions forcibly acknowledged the seropositive status for CCHFV as relatively low. Since a high effect size was identified in some reports, with most of the included seroprevalence estimates well below 10%, the reliability and generalizability of the parent studies should, therefore, be carefully evaluated, and in turn, our meta-analysis was affected by the very same shortcomings.

## 5. Conclusions

In conclusion, our study suggested that CCHFV should be considered as a far more common occurrence than previously acknowledged, particularly among workers exposed to live animals and/or their blood and bodily fluids, as well as farmers. This increased risk is not limited to middle- and low-income countries, as in European countries a moderate circulation of the pathogen in occupational settings has also been described, urging for a more accurate surveillance of clinical cases hinting at CCHFV infections, and for a more extensive promotion of appropriate PPE and preventive interventions. CCHF serves as a compelling instance of a disease ideally suited for the One Health approach. Consequently, the key to enhancing the preparedness, capacity, and response to an outbreak lies in the pivotal roles played by collaboration and networking. Occupational physicians, i.e., the medical professionals responsible for health promotion in the workplace, could be instrumental in promoting appropriate surveillance programs and improving the adherence to preventive measures in high-risk settings.

## Figures and Tables

**Figure 1 tropicalmed-08-00452-f001:**
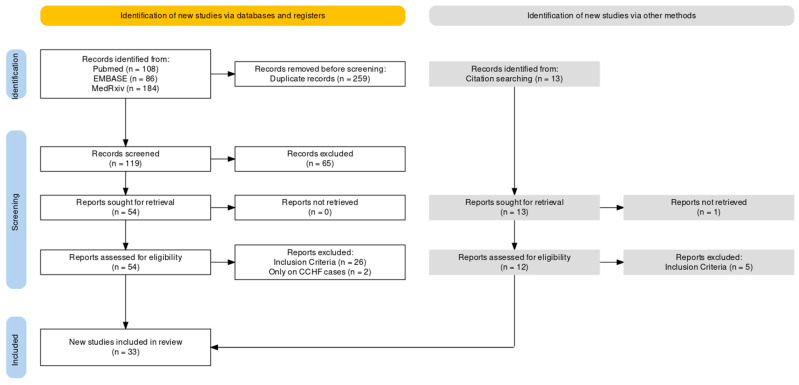
Flow chart of included studies.

**Figure 2 tropicalmed-08-00452-f002:**
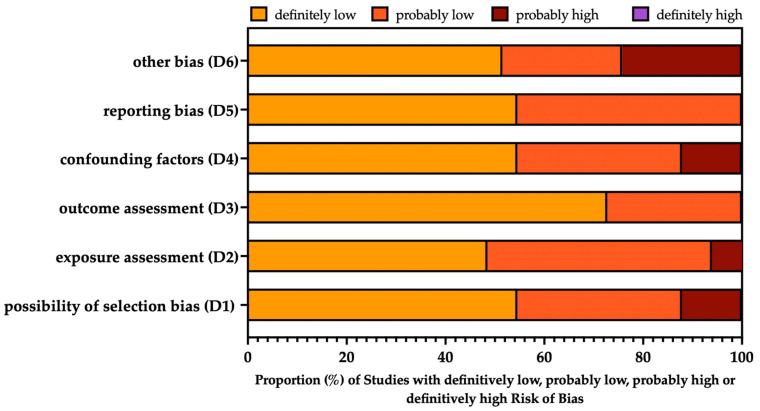
Summary of the risk of bias assessment according to the National Toxicology Program’s (NTP) Office of Health Assessment and Translation (OHAT) handbook and the respective risk of bias (ROB) tool [17,18].

**Figure 3 tropicalmed-08-00452-f003:**
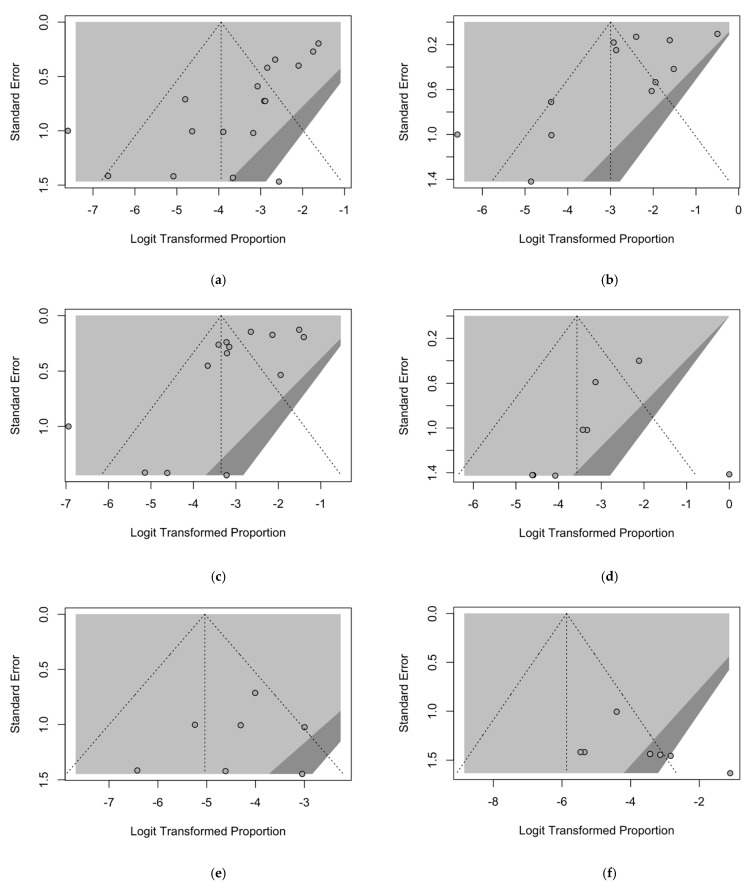
Border-enhanced funnel plots for occupational groups included in the meta-analysis: (**a**) abattoir workers, (**b**) animal handlers, (**c**) farmers, (**d**) rangers/hunters, (**e**) healthcare workers (HCWs), and (**f**) veterinarians.

**Figure 4 tropicalmed-08-00452-f004:**
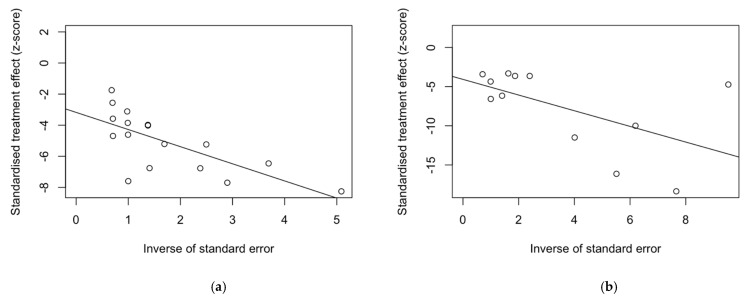
Radial plots for estimates included in the meta-analysis: (**a**) abattoir workers, (**b**) animal handlers, (**c**) farmers, (**d**) rangers/hunters, (**e**) healthcare workers (HCWs), and (**f**) veterinarians.

**Table 1 tropicalmed-08-00452-t001:** Summary of the retrieved seroprevalence studies for Crimean Congo Hemorrhagic Fever Virus: occupational settings.

Study (Year)	Country (Region)	Timeframe	Number of Samples	Occupational Settings	Cases (No.)	Positive (No., %)
Ergonul et al. (2006) [61]	Turkey (Aydin and Tokat regions)	2003	83	Veterinarians	83	1, 1.20%
Ergonul et al. (2007) [21]	Turkey (Ankara)	October 2003	75	Healthcare Workers (HCWs)	75	1, 1.33%
Andriamandimby et al. (2011) [38]	Madagascar (Nationwide)	September 2007 May 2008	1995	Abattoir Workers	1995	1, 0.05%
Xia et al. (2011) [57]	Mainland China (Yunnan)	April 2008 June 2008	1657	Farmers	318	13, 4.09%
				Animal Handlers	630	32, 5.08%
				General Population	709	12, 1.69%
Ertugrul et al. 2011 [58]	Turkey (Aydin)	n.a.	429	Farmers	167	33, 19.76%
				General Population	262	51, 19.47%
Chinikar et al. (2012) [59]	Iran (Razavi, Northern/Southern Khorasan)	2004–2005	108	Abattoir Workers	108	16, 14.81%
Hadinia et al. (2012) [39]	Iran (Yasuj)	n.a.	108	Abattoir Workers	68	3, 4.41%
Sidira et al. (2012) [62]	Greece (Nationwide)	June 2009 December 2010	1611	Farmers	756	50, 6.61%
				General Population	573	11, 1.92%
Sargianou et al. (2013) [40]	Greece (Achaia)	March 2012 July 2012	207	Abattoir Workers	39	2, 5.13%
				Animal Handlers	39	7, 17.95%
				Farmers	32	4, 12.50%
				Rangers/Hunters	29	1, 3.45%
Gozel et al. (2013) [63]	Turkey	2002–2012	190	HCWs	190	1, 0.52%
Sidira et al. (2013) [41]	Greece (Imathia)	2010–2011	277	Abattoir Workers	19	0, -
				Farmers	200	5, 2.50%
				Rangers/Hunters	32	1, 3.13%
Yagci-Caglayik et al. (2014) [60]	Turkey (provinces of Adana, Aydin, Erzurum, Gaziantep, Istanbul, Samsun, Yozgat)	n.a.	1066	Farmers	26	3, 11.54%
				Animal Handlers	38	3, 7.89%
				General Population	969	19, 1.96%
Mohd Shukri et al. (2015) [24]	Malaysia (Peninsular Malaysia)	2012–2014	85	Farmers	85	0, -
Wasfi et al. (2016) [42]	Tunisia (Nationwide)	Summer 2014	219	Abattoir Workers	38	2, 5.26%
				General Population	181	5, 2.76%
Akuffo et al. (2016) [13]	Ghana (Ashanti)	May 2011 November 2011	109	Abattoir Workers	109	6, 5.50%
Gazi et al. (2016) [43]	Turkey (Manisa)	September 2012 December 2012	450	Farmers	232	9, 3.88%
				Rangers/Hunters	29	0, -
				General Population	92	3, 3.26%
Cikman et al. (2016) [44]	Turkey (Erzinkan)	2015	372	Animal Handlers	277	46, 16.61%
Mostafavi et al. (2017) [23]	Iran (Sistan and Beluchistan)	2011	190	Abattoir Workers	188	31, 16.49%
Temocin et al. (2018) [22]	Turkey (Central Anatolia)	2016	112	HCWs	112	2, 1.79%
Vawda et al. (2018) [45]	South Africa (Free State and Northern Cape)	April 2016 February 2017	387	Abattoir Workers	245	2, 0.82%
				Animal Handlers	64	0, -
				Farmers	12	0, -
				Veterinarians	11	0, -
				Rangers/Hunters	49	0, -
Almasri et al. (2019) [14]	Saudi Arabia (Makkah)	October 2013	80	Abattoir Workers	80	0, -
Shahbazi et al. (2019) [46]	Iran (Kurdistan Province)	2014	250	Abattoir Workers	50	1, 2.00%
				HCWs	50	0, -
				Rangers/Hunters	50	0, -
				General Population	100	0, -
Mourya et al. (2019) [55]	India (Gujarat)	2015–2017	4953	Abattoir Workers	104	1, 0.96%
				Animal Handlers	723	1, 0.14%
				Farmers	1034	1, 0.10
				Veterinarians	104	0, -
				HCWs	307	0, -
				General Population	1677	0, -
Aydin et al. (2020) [47]	Turkey (Erzurum)	n.a.	91	Abattoir Workers	25	1, 4.00%
				Animal Handlers	32	4, 12.50%
				Veterinarians	15	0, -
				General Population	19	0, -
Shahid et al. (2020) [49]	Pakistan (Pubjab)	November 2016 March 2017	453	Abattoir Workers	137	9, 6.57%
				Animal Handlers	316	17, 5.38%
Arteaga et al. (2020) [50]	Spain (Castilla-Leon)	May 2017 May 2017	516	Animal Handlers	81	1, 1.23%
				Veterinarians	1	0, -
				Abattoir Workers	6	0, -
				HCWs	10	0, -
				Rangers/Hunters	2	1, 50.0%
				General Population	244	1, 0.41%
Head et al. (2020) [51]	Kazakhstan (Zhambyl)	June 2017	946	Animal Handlers	163	2, 1.23%
				Farmers	50	0, -
				Veterinarians	15	0, -
				HCWs	21	1, 4.76%
				General Population	437	8, 1.83%
Shahid et al. (2020) [48]	Pakistan (Punjab)	October 2016 May 2017	1052	Animal Handlers	468	15, 3.21%
				General Population	390	2, 0.51%
Msimang et al. (2021) [25]	South Africa (Free State and Northern Cape)	October 2017 February 2018	1040	Abattoir Workers	382	0, -
				Farmers	469	18, 3.84%
				Veterinarians	117	0, -
				Rangers/Hunters	72	3, 4.17%
Salmanzadeh et al. (2021) [52]	Iran (Ahvaz)	2020	104	Abattoir Workers	64	7, 10.94%
				Veterinarians	8	0, -
Çitil et al. (2021) [53]	Turkey (Tokat)	n.a.	2319	Animal Handlers	768	64, 8.33%
				Farmers	351	37, 10.54%
				General Population	1284	70, 5.45%
Evans et al. (2021) [61]	Myanmar (Central Myanmar)	June 2016 August 2018	102	Rangers/Hunters	65	7, 10.77%
				General Population	30	3, 10.00%
Atim et al. (2022) [54]	Uganda (Arua and Nakaseke)	2018	800	Animal Handlers	386	146, 37.82%
				Farmers	414	75, 18.12%

**Table 2 tropicalmed-08-00452-t002:** Crude estimates of Crimean Congo Hemorrhagic Fever Virus (CCHFV) seropositive status in the assessed occupational groups.

	Total (No./20,195, %)	Positive (No./876, %)	Prevalence (%)	Risk Ratio	95% Confidence Interval
General Population	6998, 34.65%	192, 21.92%	2.74%	1.000	Reference
Farmers	4558, 22.72%	260, 29.68%	5.67%	2.079	1.732; 2.496
Healthcare Workers	765, 3.79%	5, 0.57%	0.65%	0.238	0.098; 0.577
Animal Handlers	3505, 17.36%	323, 36.87%	9.22%	3.359	2.823; 3.997
Abattoir Workers	3657, 18.11%	82, 9.36%	2.24%	0.817	0.633; 1.533
Rangers/Hunters	328, 1.62%	13, 1.48%	3.96%	1.445	0.833; 2.506
Veterinarians	354, 1.75%	1, 0.11%	0.28%	0.103	0.015; 0.733

**Table 3 tropicalmed-08-00452-t003:** Results of a random-effect model meta-analysis on individual occupational groups for the seropositive status of Crimean Congo Hemorrhagic Fever Virus (CCHFV) in the assessed occupational groups (note: 95% CI = 95% confidence interval).

	Pooled Prevalence per 100 Workers (95% CI)		*τ* ^2^	I^2^ (95% CI)	Q	*p*
Farmers	3.403	(2.944; 3.932)	1.986	91.1% (86.8%; 94.0%)	146.16	<0.001
Healthcare Workers	0.644	(0.223; 1.849)	0.370	0.0% (0.0%; 70.8%)	2.41	0.867
Animal Handlers	4.751	(1.834; 11.702)	2.572	96.0% (94.4%; 97.1%)	273.80	<0.001
Abattoir Workers	1.900	(0.738; 4.808)	2.888	77.5% (64.4%; 85.8%)	71.19	<0.001
Rangers/Hunters	2.737	(0.896; 8.054)	1.011	0.0% (0.0%; 67.6%)	6.73	0.458
Veterinarians	0.283	(0.040; 1.977)	<0.001	0.0% (0.0%; 67.6%)	0.00	1.000

**Table 4 tropicalmed-08-00452-t004:** Results of a random-effect model meta-analysis on the individual pooled odds ratios (OR) for reporting a seropositive status of Crimean Congo Hemorrhagic Fever Virus (CCHFV) compared to the general population, in the assessed occupational groups (note: 95% CI = 95% confidence interval).

	Pooled Odds Ratio (95% CI)		*τ* ^2^	I^2^ (95% CI)	Q	*p*
Farmers	2.280	(1.419; 3.662)	0.238	55.8% (6.6%; 79.0%)	18.08	0.021
Healthcare Workers	3.678	(0.620; 21.835)	0.733	0.0% (0.0%; 74.6%)	0.95	0.967
Animal Handlers	2.399	(1.318; 4.369)	0.224	38.4% (0.0%; 74.1%)	9.73	0.136
Abattoir Workers	4.198	(1.060; 16.464)	0.253	5.9% (0.0%; 85.6%)	3.19	0.363
Rangers/Hunters	4.115	(0.110; 153.426)	8.380	78.6% (31.2%; 93.3%)	9.33	0.009
Veterinarians	7.966	(0.261; 242.834)	3.373	55.0% (0.0%; 89.0%)	2.22	0.136

**Table 5 tropicalmed-08-00452-t005:** Summary of the Egger’s test for funnel plot asymmetry (SE = standard error).

	t	df	Bias (SE)	Intercept (SE)	*p*-Value
Animal Handlers	−1.96	10	−4.078 (2.080)	−1.000 (0.455)	0.078
Farmers	−2.51	12	−3.471 (1.383)	−1.601 (0.340)	0.027
Abattoir Workers	−5.37	15	−3.176 (0.592)	−1.102 (0.283)	<0.001
Rangers/Hunters	−0.13	6	−1.247 (0.905)	−1.852 (0.709)	0.218
Healthcare Workers	−0.56	5	−0.880 (1.582)	−3.366 (1.658)	0.602
Veterinarians	−1.32	6	2.940 (2.222)	−7.790 (3.039)	0.234

## Data Availability

Raw data are available from the authors.

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
