# Peer review of "Seroprevalence of Crimean Congo Hemorrhagic Fever Virus in Occupational Settings: Systematic Review and Meta-Analysis"

_tropicalmed, 2023, doi:10.3390/tropicalmed8090452_

Round 1

Reviewer 1 Report

The authors should state a clear research question in the background.

In Figure 4:… “Visual inspection of contour-enhanced funnel plots suggested substantial evidence of publication bias for most of included groups, that was confirmed by Egger’s test for abattoir workers (subfigure a; p < 0.001), farmers (subfigure c, p = 0.020), veterinarians (subfigure f; p= 0.030). A likely publication bias was similarly hinted for studies on animal handlers (subfigure b, p = 0.074), while it reasonably spared HCWs (subfigure e, p = 0.590) and rangers/hunters (subfigure d, p = 0.859)” should be removed to the results section.

In Figure 5:…. “Visual inspection identified suggested that small study bias can be reasonably ruled out among abattoir workers (Figure 5a), animal handlers (Figure 5b), and HCWs (Figure 5e), while it reasonably affected all other estimates” should be removed to the results section.

There is two systematic reviews: Belobo et al (2021) dan Ergönül et al (2018), which were considered recent. Please explain what is the new? Most of your samples were included also in those reports. 

Author Response

Estimated Reviewer,

Thank you for your comments. In fact, we agreed with all of your recommendations, and we've implemented the following amendments to the main text. We are confident that the amendments we've performed substantially fulfilled the requirements from your previous review. Again, thank you for the time and the efforts you paid in order to guarantee substantial improvements to our study.

More precisely:

The authors should state a clear research question in the background.

There is two systematic reviews: Belobo et al (2021) dan Ergönül et al (2018), which were considered recent. Please explain what is the new? Most of your samples were included also in those reports. 

Thank you for this statement. In fact the previous version of our paper did not accurately addressed the aforementioned high-quality reports. Our study focused on Occupational Groups, therefore - even though some of the source reports overlap with the aforementioned reports, our study has other aims as more clearly stated by the revised PICO(s) section. More precisely:

1) background question: 

Research question was defined in accord to the “PICO” (Patient/Population/Problem; Investigated result; Control/Comparator; Outcome) strategy (Annex Table A1). More precisely: we specifically targeted groups of workers potentially exposed to Tick Bites (P) and retrieved seroprevalence rates for previous exposure to CCHF virus (i.e. CCHFV-specific IgG) (I); their estimates were compared to healthy individuals not occupationally exposed to tick and tick bites, and therefore not reasonably exposed to CCHFV (C), in order ascertain whether occupationally exposed individuals exhibited or not an increased risk for developing CCHFV infection (O).

2) regarding the significant of this study compared to previous reports:

Even though working in direct contact with infected animals and animal products has been repeatedly associated with increased risk for CCHFV infection, and despite several systematic reviews on CCHFV in humans have been published [8,12], including some estimates on the seroprevalence rate among healthcare workers [8,10,21], to the best of our knowledge no comprehensive summary of seroepidemiological studies from occupational settings has ever been published. We therefore designed the present systematic review with meta-analysis in order to provide a summary of extant literature on CCHFV seroepidemiology in occupational settings.

In Figure 4:… “Visual inspection of contour-enhanced funnel plots suggested substantial evidence of publication bias for most of included groups, that was confirmed by Egger’s test for abattoir workers (subfigure a; p < 0.001), farmers (subfigure c, p = 0.020), veterinarians (subfigure f; p= 0.030). A likely publication bias was similarly hinted for studies on animal handlers (subfigure b, p = 0.074), while it reasonably spared HCWs (subfigure e, p = 0.590) and rangers/hunters (subfigure d, p = 0.859)” should be removed to the results section.

ANSWER: The section was moved accordingly. Please note that, following an extensive revision of the searches, the text has been modified accordingly.

In Figure 5:…. “Visual inspection identified suggested that small study bias can be reasonably ruled out among abattoir workers (Figure 5a), animal handlers (Figure 5b), and HCWs (Figure 5e), while it reasonably affected all other estimates” should be removed to the results section.

ANSWER: The section was moved accordingly.

Eventually, we are confident that the improved version of our paper could cope with the high standard of TropicalMed and his qualified reviewers. Again, thank you for your suggestions.

Reviewer 2 Report

The authors should be commended for their thorough research and clear presentation of their findings. The methodology used for the systematic review and meta-analysis appears to be sound, as it adheres to established guidelines for such studies. 

Author Response

The authors would like to thank the anonymous reviewers for his/her valuable and favourable comments. 

Reviewer 3 Report

Dear Editor,

I have carefully studied the manuscript entitled "Seroprevalence of Crimean Congo Hemorrhagic Fever Virus in occupational settings: systematic review and meta-analysis" by Riccò M. et al.

The manuscript deals with a topic of increasing significance for public health worldwide, namely the epidemiology of the Crimean Congo Hemorrhagic Fever (CCHF) Virus infection. The novelty of the manuscript is the focus on occupational settings, as such a systematic review and meta-analysis has never been published before.

The manuscript is well-prepared, and the language is adequate concerning grammar and syntax. The methodology is acceptable, and the statistical approach is correct.

However, before considering publication, the authors are wellcome to discuss the following issues, aiming to ameliorate the quality of their contribution.

Major issues

1. There is an increasing concern regarding climate change worldwide. Is there any evidence of increasing prevalence and/or case fatality rate (CFR) as far as the CCHF Virus infection? The authors are wellcome to investigate a potential "dose-effect" of time on the above mentioned parameters (prevalence and CFR) and discuss their results.

2. Since there is no assess of the period covered by their literature search, the authors are awaited to explicitly state this crucial piece of information in the revised manuscript.

3. The authors are wellcome to include a sensitivity analysis investigating the effect of RoB on the outcomes studied.

4. The authors are wellcome to assess the quality of the evidence provided based on GRADE (see Guyatt GH, et al. GRADE: an emerging consensus on rating quality of evidence and strength of recommendations. BMJ. 2008;336:924-6. PMID: 18436948).

Minor issues

1. The authors are wellcome to discuss the influence of I^2 correction in case of meta-analysis including 5 or less studies, as those depicted in Figure 3 (see: von Hippel PT. The heterogeneity statistic I(2) can be biased in small meta-analyses. BMC Med Res Methodol. 2015;15:35. PMID: 25880989).

2. The authors are suggested to qualitatatively assess the outcome described in Figure A6 due to serious imprecision.

Minor editing needed.

Author Response

Estimated Reviewer 3,

Thank you so much for your recommendations. After your review, we've extensively double checked our study with the following aims: (1) implement all of your suggestion, (2) confirm whether further studies could be included through analysis of citations for removing as much as possible the bias associated with small meta-analyses.

We are confident to have fulfilled your suggestions, as reported in the following lines. Again, thank you:

1. There is an increasing concern regarding climate change worldwide. Is there any evidence of increasing prevalence and/or case fatality rate (CFR) as far as the CCHF Virus infection? The authors are wellcome to investigate a potential "dose-effect" of time on the above mentioned parameters (prevalence and CFR) and discuss their results.

ANSWER: Estimated Reviewer, as our study was focused on seroprevalence rates, its design did not included retrieval and calculation of pooled CFR (as from the systematic review from Belobo et al.). As a consequence, we're unable to provide the expertise and the insight you've recommended on this specific topic. On the other hand, we've checked whether any time trend could be ascertained or not: we've implemented the following statement in the "limits" section: 

Fourth, even though no backward limit was defined, the collected studies were performed across a very broad timespan, from 2006 to 2022, with some studies reasonably beginning sample collection in 2003-2004 [21,59], or even in the previous decades. Laboratory techniques have evolved over time, and even though ELISA and immunofluorescence assays have replaced older techniques [2], no commercial test has been approved for routine human diagnostics [2,3,104]. While the overall diagnostic performances of commonly referred tests are far from optimal (Sensitivity 80.4%, 95%CI 69.5 to 91.3; Specificity 100%) [2], some warnings have been issued about the potential sensitivity on genetic variants of CCHF, such as the strain AP92 [41]. Nonetheless, no significant time trend was eventually identified (Annex Figure A10): despite some claims about increasing seroprevalence rates because of increased testing capacity [2,8,17], our estimates were seemingly not affected by any decennial trend.

Figure A10 was specifically designed and included as annex material.

2. Since there is no assess of the period covered by their literature search, the authors are awaited to explicitly state this crucial piece of information in the revised manuscript.

The above recommendation was more extensively addressed in the aforementioned section of the main text. On the other hand, we've included the following statement:

"In the present systematic review and meta-analysis, the pooled seroprevalence for CCHFV-IgG on occupationally exposed individuals in the timeframe 2006-2022 ..."

3. The authors are wellcome to include a sensitivity analysis investigating the effect of RoB on the outcomes studied.

Sensitivity Analysis was performed as recommended: please check Figures A8+A9 and the section of results entitled: "sensitivity analysis": 

3.4 Sensitivity Analysis

Sensitivity analysis was performed by removing a single study at a time. Pooled estimates for prevalence (Annex Figure A8) were not affected in terms of residual heterogeneity, that remained substantial for animal handlers, farmers, and abattoir workers. On the contrary, when seroprevalence rates of occupational groups were compared to the general population (Annex Figure A9), estimates for animal handlers were affected by the removal of the studies from Çitil et al. [53] (OR 2.39, 95%CI 1.32 to 4.33, I2 = 9%), and Yagci-Caglayik et al. (OR 1.99, 95%CI 1.17 to 3.38, I2 = 18%) [60]. When dealing with Abattoir workers, the removal of the study from Wasfi et al. [42] led to an estimated OR equals to 9.04, 95%CI 1.40 to 58.22 with no residual heterogeneity. Interestingly, the removal of the study from Gazi et al. [43] had a similar impact on the pooled estimate for rangers and hunters (OR 243.0, 95%CI 8.13 to 7260.08), and that from Head et al. on healthcare workers (OR 54.11, 95%CI 1.51 to 1942.31) [51].  

4. The authors are wellcome to assess the quality of the evidence provided based on GRADE (see Guyatt GH, et al. GRADE: an emerging consensus on rating quality of evidence and strength of recommendations. BMJ. 2008;336:924-6. PMID: 18436948).

Even though our study was not designed in order to provide clinical recommendations, we agreed that the assessment of the quality through GRADE could improve the overall significant of this study, and therefore Annex Table A3 and A4 were implemented, as well as the following section of the results:

3.6 Summary of evidence

The level of evidence by GRADE for prevalence estimates (Annex Table A3) was moderate for abattoir workers, farmers, animal handlers, and veterinarians, due to the findings being downgraded due to quality of source studies, with low evidence for estimates on rangers/hunters and healthcare workers. When taking in account corresponding estimates on the general population (Annex Table A4), the evidence was again moderate for animal handlers and farmers, and also for healthcare workers, while it was low for abattoir workers, and very low for rangers/hunters and veterinarians due to sample size and quality issues.

1. The authors are wellcome to discuss the influence of I^2 correction in case of meta-analysis including 5 or less studies, as those depicted in Figure 3 (see: von Hippel PT. The heterogeneity statistic I(2) can be biased in small meta-analyses. BMC Med Res Methodol. 2015;15:35. PMID: 25880989).

ANSWER: we thank you the Rev.3 for this specific recommendation. Results were extensively revised in order to implement the reporting of I2 with their 95%CI as recommended by von Hippel (please refer to Table 3 and 4). However, please also note that estimates benefited for analysis of citations from included papers, that increased to number of included studies from 25 to 33.

2. The authors are suggested to qualitatatively assess the outcome described in Figure A6 due to serious imprecision.

ANSWER: Estimated Rev.3, thank you for your recommendation. In fact, previous design of Figure A6 was scarcely informative because of its serious imprecision due to the lack of samples. However, following the second round of searches, the imprecision has been significantly reduced. However, main text has been modified accordingly.

Again, thank you for the time you've spent in the accurate assessing of our study, and for the recommendations that have significantly improved the significance of this report.

(regarding the editing of the main text, please note that the paper has been radically revised)

Reviewer 4 Report

The presented systematic review and meta-analysis concerns a disease that is often underestimated. Therefore, I think that the topic is current and relevant. The authors present sufficiently extensively the studies and publications on the given topic. Appropriate statistical methods were used to analyze the data provided. The information is clearly presented, and the results correspond to the set goals. The writing style is relevant to the scientific information presented.

Author Response

The authors would like to thank the anonymous reviewers for his/her favourable and valuable comments

Reviewer 5 Report

Crimean-Congo hemorrhagic fever virus (CCHFV) is an emerging tick-borne virus that is responsible for a fatal human disease. The majority of CCHFV infections is asymptomatic or cause mild clinical signs. The presented data demonstrated that CCHFV infection may represent a far more common occurrence, even in countries not usually associated with CCHF. The results of manuscript are of substantial interest for Public Health professionals. The article is well written, and the methodology used is adequate. In my opinion, this study has several limits, but the authors describe these limits in detail. The article is interesting and worthy of publication. 

Author Response

(The authors gave the same response as above.)

Round 2

Reviewer 3 Report

I have studied the revised version of the manuscript entitled "Seroprevalence of Crimean Congo Hemorrhagic Fever Virus in occupational settings: systematic review and meta-analysis".

The authors have successfully responded to every single query raised. The quality of the manuscript has been substantially ameliorated. There are no additional queries/issues.